# Ultrasonic Freezing Reduces Protein Oxidation and Myofibrillar Gel Quality Loss of Common Carp (*Cyprinus carpio*) during Long-Time Frozen Storage

**DOI:** 10.3390/foods10030629

**Published:** 2021-03-16

**Authors:** Qinxiu Sun, Baohua Kong, Shucheng Liu, Ouyang Zheng, Chao Zhang

**Affiliations:** 1College of Food Science and Technology, Guangdong Ocean University, Zhanjiang 524088, China; sunqinxiugo@163.com (Q.S.); Lsc771017@163.com (S.L.); zhengouyang07@163.com (O.Z.); 2College of Food Science, Northeast Agricultural University, Harbin 150030, China; zhangchaocheer@163.com

**Keywords:** ultrasonic freezing, protein oxidation, gel property, frozen storage, common carp

## Abstract

Ultrasonic freezing (UF) is an effective method to increase the freezing speed and improve the quality of frozen food. The effect of UF on myofibrillar protein oxidation and gel properties of common carp (*Cyprinus carpio*) during frozen storage were investigated with air freezing (AF) and immersion freezing (IF) as controls. The results showed that the carbonyl and dityrosine content of UF samples were lower and the free amine content was higher than those of AF and IF samples during frozen storage indicating that UF inhibited protein oxidation caused by frozen storage. The particle size of UF myofibrillar protein was the smallest among all the groups indicating that UF inhibited the protein aggregation. The UF sample had higher *G*’, *G*” value, gel strength and gel water holding capacity than AF and IF groups showing that UF reduced the loss of protein gel properties. The gel microstructure showed that UF protein gel was characterized by smaller and finer pores than other samples, which further proves that UF inhibited loss of gel properties during frozen storage. The UF sample had shorter *T*_2_ transition time than other samples demonstrating that UF decreased the mobility of water. In general, UF is an effective method to reduce protein oxidation and gel properties loss caused by frozen storage.

## 1. Introduction

Common carp (*Cyprinus carpio*) is the most important freshwater fish in the world, especially in China. There were about 3.01 million tons of carp were farmed in 2019, accounting for about 70% of total freshwater fish production in China. It is rich in protein, unsaturated fatty acids and trace elements. They are a common food that easily spoilage and deteriorates; therefore, measures should be taken to better preserve fish [1]. Freezing is a common method for the preservation of aquatic products but oxidation of proteins and lipids still occur during frozen storage resulting in flavor deterioration and texture damage of aquatic products [2]. Freezing speed directly affects the quality of frozen food. Slow freezing forms large and irregular ice crystals that can destroy the muscle tissue of fish [3]. Fast freezing may decrease the size of ice crystals and increase freezing speed; thus, fast freezing technology in food is favored by researchers [4].

Ultrasonic freezing (UF) is a novel fast freezing technique developed in recent years. Studies have shown that it can effectively accelerate the freezing process, reduce the ice crystals size and improve the quality of frozen foods. The mechanisms may be as follows: On the one hand, ultrasonic waves can produce cavitation bubbles, which can be used as crystal nuclei to promote the formation of ice crystals. On the other hand, micro-jet generated by ultrasonic wave can break large ice crystals into small ice crystals, which can be used as crystal nuclei to promote recrystallization of ice crystals [5,6]. At present, UF has been used in many frozen food including vegetables, fruits, dough and meat products [5,6,7,8]. However, most studies focused on the effect of UF on the freezing speed and quality of frozen food [9]; only a few studies studied the effect of UF on food quality during long-term frozen storage [10]. The growth of different forms of ice crystals is different during frozen storage. In addition, the production of reactive oxygen species during the UF process may also impact the quality of frozen food, which may not be significant in the short-term frozen process. The storage period of frozen fish was about 90–180 d; the protein and quality of fish changed significantly after long term frozen storage (180 d). Therefore, it is necessary to explore whether UF can continue to maintain its advantages during long-term frozen storage.

We have previously studied the effects of different power of UF on the freezing speed and quality of common carp (*Cyprinus carpio*). The results showed that 175 W of UF could accelerate the freezing progress and improve the muscle quality of fish [11]. Myofibrillar protein (MP) is the main protein in fish; it is prone to oxidation during frozen storage, which leads to protein denaturation and affects product quality. The gel properties of MP determine the quality properties of fish products and is an important factor affecting the sensory and functional properties of meat. Therefore, the effects of UF on the protein oxidation and gel properties of MP of common carp during frozen storage was investigated with air freezing (AF) and immersion freezing (IF) as controls.

## 2. Materials and Methods

### 2.1. Chemicals

The 2,4-dinitrophenylhydrazine (DNPH), piperazine-N,N’-bis-2-ethanesulphonic acid (PIPES), sodium monohydrogen phosphate, ortho-phthaldialdehyde (OPA), hydrochloric acid, ethanol, methanol, sodium dodecyl sulfate (SDS), guanidine hydrochloride, ethyl acetate, trichloroacetic acid (TCA), β-mercaptoethanol, 1-anilinonaphathalene-8-sulfonate (ANS), L-leucine, disodium hydrogen phosphate, resin and glutaraldehyde were purchased from the Suzhou Yacoo Science Co., Ltd. (Suzhou, China).

### 2.2. Frozen Fish

Non-pregnant common carp were purchased from the local aquatic products market (Harbin) with an average live weight of 1100 ± 60 g. Then, they were cut into pieces (about 5 cm in length) perpendicular to the fish body (each piece weighed about 210 ± 15 g). Each fish piece was washed with running water and packed separately in a polyethylene zipper bag. The samples were then frozen by AF, IF, or UF. For AF, the sample was frozen at -25.0 °C ± 0.5 °C in an ordinary refrigerator until the central temperature of the fish samples reached −18 °C. For IF and UF, the samples were placed in a fixed position inside the freezing tank of the ultrasonic freezer (Nanjing Xianou Co., Ltd., Nanjing, China). The temperature of the coolant (95% ethanol) was kept constant (−25 °C ± 0.5 °C) and the device output power was set to 0 (IF) and 175 W (UF) at a frequency of 30 kHz. The ultrasound was performed for 9 min when the central temperature of the fish piece dropped to 0 °C in ultrasonic mode 30 s/on-30 s/off cycles. After the geometric center temperature of the sample reached −18 °C, all samples were transformed to a −18 °C ± 1 °C refrigerator and stored for 180 d. Samples were taken for analysis on day 0, 30, 60, 90 and 180.

### 2.3. Extraction of Fish MP

MPs were extracted according the method described by Sun, Chen, Xia, Kong and Diao [12]. The frozen samples were thawed in a refrigerator (4 °C) to a central temperature of 4 °C. Then, the fish meat was removed from the fish piece and minced. Minced fish meat samples (60 g) were mixed with 240 mL phosphate buffer (PBS, 20 mm, pH 7.0, 0.1 M NaCl) and then homogenized in a tissue masher (Shanghai Zuole Instrument Co., Ltd., Shanghai, China) for 1 min. The mixture was centrifuged at 6500× *g*, 4 °C for 15 min with a refrigerated centrifuge (Shanghai Lu Xiangyi Instrument Co. Ltd., Shanghai, China) and the pellet was extracted twice with 240 mL PBS according to the above steps. The precipitate was then redissolved with 240 mL of NaCl (0.1 M) and homogenized in a tissue masher for 1 min. The mixture was then centrifuged at 6500× *g*, 4 °C for 15 min and the pellet was repeatedly washed again according to the above steps. The precipitate was then redissolved with 240 mL NaCl (0.1 M) and homogenized in the tissue masher for 1 min and then filtered through four layers of gauze. After filtration, the suspension was centrifuged at 6500× *g*, 4 °C for 15 min to obtain a MP pellet. The concentration of protein was measured with the Coomassie bright blue method.

### 2.4. Carbonyl Content

The carbonyl content of fish MP was measured as described by Zhang, Fang, Hao and Zhang [13]. Briefly, 1 mL MP solution (2 mg/mL) was reacted with 10 mm DNPH (25 °C) in the dark for 1 h. The sample was swirled every 10 min and allowed to stand for 10 min after addition of 20% TCA. The solution was centrifuged at 10000× *g,* 4 °C for 5 min. To remove the excess DNPH, the precipitate was washed three times with 1 mL washing solution (ethanol: ethyl acetate, 1:1, *V*/*V*). The precipitate was reacted with 3 mL guanidine hydrochloride (6 moL/L) for 15 min at 37 °C. The absorbance of the solution was then measured at 370 nm and the carbonyl content (nmoL/mg) was calculated using the absorption coefficient of protein hydrazone 22,000 M^−1^ cm^−1^.

### 2.5. Dityrosine Content

The dityrosine content of the fish MP was measured as described by Zheng, Zhou, Zhang, Wang and Wang [14]. Briefly, the fluorescence intensity of the MP solution (about 1 mg/mL) was measured by a fluorescence spectrophotometer (F-4500, Hitachi, Tokyo, Japan) with excitation at 325 nm and emission at 410–430 nm. The corrected fluorescence was calculated by dividing the fluorescence intensity by the actual MP concentration (mg/mL).

### 2.6. Free Amine Content

The free amine content of the sample was conducted as described by Sun et al. [10]. The OPA solution was prepared containing 40 mg OPA, 1 mL methanol, 1.095 g borax, 50 mg SDS and 100 μL β-mercaptoethanol and stabilized with purified water to 50 mL. Next, 4 mL OPA solution was reacted with 200 μL MP solution (3 mg/mL). After full mixing, the mixture was stored at 35 °C for 2 min. The solution absorbance at 340 nm was then measured. The free amine content of the sample was calculated using a 0.25–2.00 mm L-leucine calibration curve as the standard.

### 2.7. Surface Hydrophobicity

The surface hydrophobicity of the sample was measured as described by An, You, Xiong and Yin [15]. MP was adjusted to 0.08, 0.16, 0.24, 0.32 and 0.4 mg/mL with a final volume of 4 mL. The protein solution was mixed with 40 μL ANS (8.0 mM). After reacting for 20 min, at 25 °C, the relative fluorescence intensity of the ANS-protein conjugate was determined with a F-4500 fluorescence spectrophotometer (F-4500, Hitachi, Tokyo, Japan). The excitation was set to 390 nm and the emission was set to 470 nm. Surface hydrophobicity is the initial slope of the curve of fluorescence intensity and protein concentration (mg/mL).

### 2.8. Particle Size

The particle size of the fish MP was determined by a Mastersizer (Mastersizer 2000, Malvern Instruments Ltd., Malvern, UK). Ten milliliters of protein solution (15 mg/mL) was slowly added to 500 mL distilled water and then stirred until the uniformity index of the protein solutions ranged from 0.538 to 0.622 indicating good dispersion. The particle size was expressed as volume average diameter (*D*_43_).

### 2.9. Rheological Properties

The rheological properties of the samples were determined by a Model VOR rheometer (Bohlin Rheology, Lund, Sweden). A 40 mg/mL MP solution was added between two parallel plates (50 mm in diameter) spaced 1 mm apart. Adopt the temperature sweep test, set the strain of 1%, the oscillation frequency of 0.1 Hz and gradually heat from 30 °C to 80 °C at a heating rate of 1 °C/min. The *G*’ (storage modulus) and *G*” (loss modulus) of the samples were used to reflect the change of rheological properties of MP.

### 2.10. The Water Holding Capacity of Fish MP Gel

Preparation of fish MP gel was as follows. First, 40 mg/mL MP solution (20 mL) was added into the weighing bottle. The protein gel was prepared by two-step heating at 45 °C for 15 min and then at 80 °C for 15 min. The samples were subsequently cooled to room temperature with ice and stored in a 4 °C refrigerator for 12 h.

The water holding capacity (WHC) of protein gel was determined by a centrifugal method. Briefly, approximately 5 g (*W*_0_) protein gel was added to a 10 mL centrifuge tube. Then, the protein gel was centrifuged in a frozen centrifuge (4 °C, 10,000× *g*) for 20 min. After centrifugation, the water on gel surface was carefully blotted off with filter paper, then the mass of the gel was accurately weighed (*W*_1_). The WHC of protein gel was calculated as following:(1)$$\mathrm{WHC}\;(\%)\;=\;\frac{W_{0}-\;W_{1}}{W_{0}}\;\times \;100\%$$

### 2.11. The Gel Strength of Fish MP

The gel strength of the samples was determined using a texture analyzer (TMS-Pro, FTC, Sterling, VA, USA) with a P/0.5 probe. Before testing, the protein gel was balanced at room temperature (25 °C) for 1 h. The puncture mode was used to measure the gel strength of the sample. The penetration rate was 1 mm/s and the force arm was 5 kg. The gel strength was expressed by the maximum penetration force.

### 2.12. Low Field Nuclear Magnetic (LF-NMR)

The water distribution in protein gel were analyzed by a Minispec mq20 LF-NMR analyzer (Bruker Optik GmbH Ettlingen, Ettlingen, German). The fish MP gel was prepared according to the above method. A glass tube was used instead of the weighing bottle to make a gel with a height of 4 cm. The prepared gel was equilibrated at 25 °C for 1 h and inserted into the magnetic pulse NMR analyzer. The analyzer operated at 25 °C and the resonance frequency was 19 MHz. The spectral width was set to 200 kHz and the receiving gain was 20 dB. The transverse relaxation curve was derived using the Carr-Purcell-Meiboom-Gill pulse sequence and the CONTIN algorithm.

### 2.13. The Microstructure Fish MP Gel

The microstructure of fish MP gel was observed using scanning electron microscopy (SEM) (S3400, Hitachi, Tokyo, Japan) according to Sun, Zhang, Li, Xia and Kong [16]. Briefly, protein gel samples (1 × 2 × 3 mm^3^) were fixed in 0.2 M PBS (pH 7.2), treated with 2.5% glutaraldehyde for 6 h at 4 °C and the samples were washed with 0.2 M PBS (pH 7.2) three times and washed with successive solvents composed with an increasing content of ethanol (50%, 70%, 80%, 90% and 100%, respectively). The samples were lyophilized and glued to conductive resin followed by spraying with gold/palladium coating before SEM imaging.

### 2.14. Statistical Analysis

Three batches of samples were made in this study. All measurements were performed three times for each batch of samples. Data were analyzed with the general linear model program of Statistix 8.1 software (Analytical Software, St. Paul, MN, USA). The difference among different treatments was measured by one-way analysis of variance followed by Tukey’s multiple comparisons with 95% confidence intervals (*P* < 0.05).

## 3. Results and Discussion

### 3.1. Change in Carbonyl Content

The degree of protein oxidation was evaluated by measuring the content of protein carbonyl, which is an important evaluation index of muscle protein oxidation [13]. Figure 1 shows no obvious distinction in the carbonyl contents of samples among all groups at the initial stage of storage (0 day) (*P* > 0.5). At longer storage time, the carbonyl content of all the samples increased to different degrees. After frozen storage for 180 d, the carbonyl content of AF sample increased from the original 1.71 nmoL/mg protein to 7.73 nmoL/mg protein, which was 6.92% and 22.50% higher than that of IF and UF samples, respectively (*P* < 0.05). Utrera, Armenteros, Ventanas, Solano and Estévez [17] reported that the side chains of protein lysine, arginine, or proline would be oxidized to carbonyl under the action of transition metals (such as iron) and reactive oxygen species. Frozen storage can accelerate the pro-oxidation of iron and other oxidants. This is mainly because the freezing of free water leads to a decrease in water activity and an increase in solute concentration in the components of unfrozen water, which leads to more effective collisions between reactants [18]. As a result, the protein attaches to a large amount of unfrozen water and pro-oxidant concentrates in the water and strengthens the pro-oxidant effects. In addition, damage to the ultrastructure of muscle cells during freezing would accelerate the release of erythroid iron and other oxidants, which might promote the production of carbonyl groups [19]. The increase in carbonyl content during frozen storage was effectively inhibited by UF treatment, which may be because that UF maximally preserved the integrity of fish muscle cells [10], reduced the release of oxidase in cells and thus inhibited the oxidation reaction during frozen storage. In addition, protein oxidation may be associated with other oxidative events and factors such as oxidized lipids, oxidases, free radicals and heme pigments [20].

The oxidation of one meat ingredient results in the formation of chemicals that accelerate the oxidation of other components. For example, malondialdehyde is a product of lipid peroxidation and can react with protein to form carbonyl groups [21]. Our previous study showed that the degree of fat oxidation (thiobarbituric acid reactive substances value) in UF samples was significantly lower than that in AF and IF samples [10], which may further explain the lower carbonyl content in UF samples than in other samples. The formation of carbonyl groups led to a change in the protein structure (such as expansion or aggregation of protein) as well as the reduction of solubility, emulsification and gel properties, which reduced the further processing properties of the protein [22]. In addition, the formation of carbonyl groups during frozen storage can also reduce muscle quality including moisture holding capacity, texture properties, nutritional value and flavor of samples [23]. Therefore, the use of UF to freeze fish inhibited the oxidation of protein during frozen storage and reduced its quality loss.

### 3.2. Change in Free Amine Content

The free amine content of MP is often used to provide valuable information of lysine side chains [24]. There was no significant difference between the free amine content of AF, IF and UF at day 0 of storage (*P* > 0.05), which may be because the freezing time was short and the oxidation degree of the frozen samples was too small to cause significant differences. With the extension of storage time, the free amine content of all samples decreased gradually and the decrease rate of AF samples was the fastest (*P* < 0.05). After 180 days of frozen storage, the AF sample had the lowest free amine content—It was 6.58% and 13.89% lower than IF and UF samples, respectively. The ε-NH_2_ lysine residue is unstable and easily generates carbonyl groups through a deamination process during oxidation. The carbonyl derivatives can react with amino groups resulting in the reduction of free amine content [25]. In addition, 1% SDS was added in the determination of free amine content to destroy non-covalent protein-protein interactions. Therefore, the covalent addition of oxido-quinone and protein free amine might be another reason for the loss of ε-NH_2_. Zhang et al. [24] also found that frozen storage decreased the free amine content of pork MP. The free amine content of UF sample was the highest among all groups, which was associated with the lowest oxidation of the UF samples—this result corresponded to the carbonyl content.

### 3.3. Change in Dityrosine Content

Dityrosine is a macromolecular amino acid formed by the C-C coupling of two L-tyrosine after oxidation with a characteristic fluorescence in the range of 400–420 nm. Due to its stability and fluorescence properties, dityrosine is often used as a biomarker for oxidative modification of protein side chains [26]. Figure 2 shows that there was no obvious difference in the dityrosine content among all groups on day 0 of storage (*P* > 0.05). This indicates that at the initial stage of frozen storage, no significant oxidation reaction occurred in any group because the freezing time was so short. The dityrosine content of all groups increased significantly with longer storage times. After 180 days of frozen storage, the dityrosine content of AF, IF and UF samples increased by 121.04%, 100.00% and 78.31%, respectively, versus at day 0 of storage. As storage time increased, muscle proteins underwent oxidation leading to the formation of hydroperoxides and carbonyl groups. The increase degree of oxidation leads to degradation of reactive sulfhydryl groups and the formation of dityrosine allowing cross-linking within or between protein molecules [27]. Colombo et al. [28] also found that the dityrosine content of proteins increased with increasing protein oxidation. Ragnarsson and Regenstein [29] suggested that muscle proteins were cross-linked by disulfide and non-disulfide covalent bonds during frozen storage, which promoted the formation of high molecular weight polymers and aggregates. Here, the increase in dityrosine content in carp samples during frozen storage was significantly inhibited by UF treatment suggesting that UF could reduce some protein cross-links such as carbonylamine interactions, thus reducing the generation of dityrosine.

### 3.4. Change in Surface Hydrophobicity

Surface hydrophobicity is often used to assess the extent of protein molecule unfolding caused by physical treatments [30]. As illustrated in Figure 2, the surface hydrophobicity of the AF sample was 286.0 So-ANS at day 0; no obvious difference among all the groups was observed (*P* > 0.05), which showed that the three different freezing conditions did not significantly affect the surface hydrophobicity of proteins. With increasing storage time, the surface hydrophobicity in all treatment groups increased. After storage for 180 days, the surface hydrophobicity of AF samples increased to 481.8 So-ANS, which was significantly higher than that of IF and UF samples (*P* < 0.05). This can be explained by the faster freezing speed of IF than AF. The ice crystals generated in this process were smaller and more uniform than AF. Therefore, the growth rate of ice crystals in IF was lower than AF during frozen storage stage. Damage to muscle structure by ice crystals can lead to outflow of cell contents (e.g., water, endogenous proteases), which changes the environment in which muscle proteins are located and thus accelerates protein denaturation. Thus, the destruction of IF muscle tissues was smaller than that of AF samples and the corresponding unfolding degree of protein of IF muscle was lower than that of AF samples. The unfolding of protein can promote the exposure of hydrophobic amino acids and increasing its surface hydrophobicity [31]. Wagner and Añón [32] found that slow freezing more easily caused protein denaturation leading to partial unfolding of the protein structure. Exposure of some hydrophobic groups located in the interior of the protein led to the increase of surface hydrophobicity. Therefore, rapid freezing can effectively inhibit the increase in surface hydrophobic during frozen storage. In addition, the increase of hydrophobicity during the frozen storage of proteins may be related to the exposure of aromatic amino acids and hydrophobic aliphatic [33]. Resendiz-Vazquez et al. [34] showed that ultrasound could unfold protein molecules exposing hydrophobic groups in the molecular interior to the polar environment and thus, increasing the surface hydrophobicity. One possible reason for the contrary conclusion in this study is that the ultrasonic power applied in this study was low (175 W). In addition, in the present study, the ultrasound was used during the freezing process. The cavitation bubbles generated by ultrasound can act as crystal nuclei to promote the formation of ice crystals. In addition, the micro-jet produced by ultrasound can break the large ice crystals into small ice crystals, which can be used as crystal nuclei and promote the recrystallization of ice crystals [9,10]. Therefore, UF can promote the generation of fine and uniform ice crystals and inhibit the protein denaturation produced by freezing. Zhao, Dong, Li, Kong and Liu [35] thought that hydrophobic interactions may occur between exposed hydrophobic residues causing aggregation of the protein. Thus, it is necessary to inhibit the increase of surface hydrophobicity of freezing food.

### 3.5. Particle Size Distribution

The distribution of protein particle size may evaluate the aggregation or unfolding of protein. The change in particle size of the protein will affect the solubility, emulsification and functional properties of protein. Here, granulometer was used to measure the changes in the average particle size of proteins during frozen storage. Figure 3 shows a large peak in the size distribution of all samples. With increasing storage time, the peak moved to the right, i.e., the size of particles increases. This change indicates that soluble proteins were rearranged into larger aggregates during frozen storage.

To better understand the changes in protein particle size, we performed statistical analysis on the average protein volume size (*D*_43_) (Table 1) of all the samples. At the beginning of storage (0 day), the *D*_43_ of AF group was significantly greater than that of IF and UF samples (*P* < 0.05). This might be due to the mechanical damage to the protein caused by the large ice crystals formed by AF during freezing process. Part of the protein structure unfolded, some subunits were exposed, some proteins dissociated and other proteins reassembled to form aggregates. After storage for 180 days, the *D*_43_ value of AF was significantly increased and was 58.56% higher than that at the beginning of storage. The growth rate of *D*_43_ in UF and IF was obviously lower than that of AF. At the end of storage, the *D*_43_ of IF and UF samples were 3.47% and 15.11% lower than that of AF samples, respectively. The increase of protein particle size during frozen storage might be related to the growth of ice crystals, which further underscored the destruction of protein structure. It exposed more active subunits, rearranged protein molecules and formed larger aggregates. In addition, oxidation of some active groups of proteins occurred during storage and cross-links (such as dityrosine) occurred between nascent groups resulting in crosslinking within or between protein molecules, which also made the protein particle size larger [36]. The UF samples formed relatively small and uniform ice crystals during freezing and the growth degree of ice crystals during frozen storage was lower than that of IF and AF samples. Therefore, UF effectively inhibited the increase of protein particle size of fish MP during frozen storage.

### 3.6. Rheological Properties

The viscoelastic properties of protein gel are a major index that determine the final quality of fish. Changes in *G*’ and *G*’’ of fish MP were monitored to investigate changes in gel viscoelastic properties of fish MP during frozen storage. The *G*’ reflects the elastic solid-state behavior of the sample and the *G*’’ measures the viscous response of the sample [37]. As shown in Figure 4, the slow increase in *G*’ at the initial stage of gel (30–39 °C) indicated that the protein structure changes were small at this stage. Then, *G*’ increased rapidly and reached the maximum value at 46 °C, indicating the onset of a gel network. The change in this stage may be mainly due to the degeneration of S_1_ subunit segment of the myosin head, which led to the binding of myosin filaments, thus, increasing the elasticity of the gel [38]. In addition, the denaturation of actomyosin at this stage also promoted protein-protein interactions and further accelerated protein-protein aggregation. The *G*’ of UF sample at 46 °C was higher than that of AF and IF samples, indicating that the myosin and actomyosin of UF were more complete and the protein thermal stability of UF was higher than those of AF and IF. Subsequently, the *G*’ decreased rapidly which was known as the gel weakening stage. This change may be due to the transformation of the helical structure of the myosin tail into a curled structure, which led to the destruction of the existing protein network structures [39]. The further increase in *G*’ after 49 °C was called the strengthening stage of the gel. This change may be due to the fact that most of the myosin molecules may unfold and transform into haptic coiled structures, increasing the number of cross-links between protein aggregates.

Figure 4 shows that both storage time and freezing condition have effects on protein *G*’ value. With extension of storage time, the *G*’ value of all the group decreased and the first peak value of AF sample decreased from 208.93 Pa at day 0 to 64.50 Pa at day 180. The *G*’ of UF samples was always the largest among the three groups during the frozen storage. Xuan, Zhang, Zhao, Zheng, Jiang and Zhong [40] suggested that the water in food recrystallized with longer storage time and the size of ice crystals became larger, which damaged the protein network and even altered the chemical bonds, thus reducing the *G*’ value of food. In addition, changes in some molecular interactions during frozen storage may also be responsible for the decrease of *G*’ value. For example, oxidation transforms sulfhydryl group into disulfide bond, breaks hydrogen bonds and exposes hydrophobic groups causing unfolding of protein structures, etc. These can all lead to a decrease in MP *G*’ [41]. The degree of protein oxidation and protein denaturation during storage of UF samples were lower than this of IF and AF samples and thus had higher values of *G*’, which corresponded to the results of protein carbonyl and dityrosine content.

The loss modulus (*G*”) is the energy dissipated in the form of heat and therefore represents the viscous property. The change of *G*” was similar to that of *G*’, but the value of *G*” was always lower than *G*’ indicating that the elasticity of protein gel was dominant. With longer storage time, the *G*” of all the groups decreased gradually. The *G*” of UF sample was the highest followed by IF and AF samples. The main reason for the decrease of *G*” of the fish protein was that protein denatures to different degrees during frozen storage, which led to a decrease in the viscosity of protein gel.

### 3.7. Water Distribution in Fish Protein Gel

LF-NMR is a nondestructive technique used to monitor the distribution of water in food [16]. Figure 5 shows four peaks in the LF-NMR curves of all samples. *T*_2b1_ and *T*_2b2_ represent the water tightly bound to the protein, *T*_21_ is the immobilized water within the protein network structure and *T*_22_ represents the free water in the gel network structure. Water with longer relaxation times are more loosely bound to macromolecules and has stronger fluidity [42]. With longer storage times, the relaxation time of all groups increased, which represents an increase in water fluidity. Figure 3 shows that the changes in *T*_2b1_ and *T*_2b2_ relaxation times were similar. At initial storage, there was no significant difference in *T*_2b1_ and *T*_2b2_ relaxation time among all the groups (*P* > 0.05). This may be because bound water is tightly bound with protein macromolecule and general heating and freezing treatments cannot significantly affect the bound water. The *T*_2b1_ and *T*_2b2_ relaxation times in all groups increased gradually with storage time and those of AF samples increased the fastest. At the end of storage, the *T*_2b1_ and *T*_2b2_ relaxation time of the AF samples increased by 64.79% and 56.24%, respectively, versus the value at initial storage. The relaxation time of *T*_2b1_ and *T*_2b2_ of UF samples increased at the slowest rate. At the end of storage, the *T*_2b1_ and *T*_2b2_ relaxation time of UF samples were 28.89% and 22.1% lower than those of AF samples, respectively. The increase of *T*_2b1_ and *T*_2b2_ during storage may be due to unfolding of the protein structure with exposure of partially hydrophobic groups leading to a reduction in the ability of the protein to bind water and thus increasing mobility of the bound water.

The UF samples had fewer surface hydrophobic groups than IF and AF samples during storage. Thus, the water mobility of UF samples was smaller than those of AF and IF samples. In addition, the decrease of *T*_2b_ relaxation time may also be related to the protein oxidation, which caused partial denaturation of the protein and reduced the binding capacity of protein molecules to water.

Immobile water was the main form of water in protein gel. On day 0 of storage, the *T*_21_ relaxation time of AF, IF and UF samples were 415.07, 392.41 and 374.72 ms, respectively. *T*_21_ relaxation time of UF samples was significantly shorter than that of AF and IF, which was related to the smaller degree of denaturation of UF samples during freezing. Our previous results showed that UF protein had high thermal stability than any other samples [10], which proved that the protein denaturation degree of UF was lower. Therefore, the gel formed by UF fish protein had a fine network structure, which could maintain more immobile water. The *T*_21_ of all the samples increased continuously with prolonged storage time. At the end of storage, the *T*_21_ relaxation time of AF samples was the longest followed by IF and UF samples. Alinovi, Corredig, Mucchetti and Carini [43] also found that frozen storage increased the relaxation time of frozen food. The increase of *T*_21_ during frozen storage may be because of the growth of ice crystals during storage [14] and an unfolded protein structure that exposed more hydrophobic groups and decreased the protein solubility, which was not conducive to the formation of a good gel structure of protein. Moreover, protein oxidation led to the formation of large protein aggregates making them form a loose gel network, thus reducing the ability of the gel to bind water.

Fine ice crystals were formed during UF and thus the extent of ice crystal growth in UF was also less than that of AF and IF samples (large and irregular ice crystals) [10], which inhibited the unfolding and oxidation of protein caused by frozen storage; therefore, UF effectively inhibited the increased of *T*_21_ relaxation time in protein gel during frozen storage.

The changes of *T*_22_ relaxation time were similar to those of *T*_21_. At the initial of storage, the *T*_22_ relaxation time of AF sample was significantly higher (*P* < 0.05) than those of the other two groups and the prolonged storage time resulted in the increase of *T*_22_ in all groups. UF effectively improved the binding capacity of gels to free water. Li et al. [2] also found that MP denaturation due to long-term frozen storage altered hydration and increased *T*_2_ relaxation time in common carp during frozen storage.

As shown in Figure 5, with the extension of storage time, the amplitude of water decreased, which may be due to the protein denaturation caused by frozen storage, resulting in the decrease of protein water binding ability. Similarly, Wang et al. [44] also found that the amplitude of gluten protein decreased with the prolonging of frozen storage time. As shown in Table 2, for all the samples, the *A*_2b1_ and *A*_2b2_ gradually decreased with the prolongation of frozen storage time, which may be due to the denaturation of proteins caused by frozen storage, reducing the ability of proteins to bind to water. The denaturation of proteins led to the formation of a poor gel, so the ability of the gel to bind to immobilized water was reduced and part of the immobilized water was converted into free water. Corresponding to this, for all the samples, the *A*_21_ decreased and the *A*_22_ increased with the prolongation of storage time. For the same storage time, the *A*_2b1_, *A*_2b2_ and *A*_21_ of UF sample were larger than those of AF and IF sample, indicating that the denaturation degree of UF MP was lower than that of AF and IF during the frozen storage.

### 3.8. Gel Strength

Protein gel strength is an important index that reflects gel quality. As shown in Figure 6A, the gel strength of UF sample (0.78 N) was obviously higher than that of AF and IF (0.60 and 0.65 N) (*P* < 0.05) at day 0 of storage. This may be attributed to the large ice crystals formed during the slow freezing process and dehydration induced protein denaturation, which prevented the protein from forming a good gel network structure [45]. The gel strength of all groups reduced with the extension of storage time. At the end of storage, the gel strength of the UF sample remained the highest followed by AF and IF samples. The quality of protein gel is closely related to the structural integrity of the protein especially the myosin integrity, which is the prerequisite for protein gelation [46]. During frozen storage, protein denaturation and aggregation were caused by disulfide bonds, hydrophobic interactions and lipid oxidation, which may hinder the ordered interactions of active functional groups between proteins. Therefore, the formation of dense and uniform gel networks was inhibited and the gel strength of the samples was reduced. The decrease in protein gel strength of fish MP by frozen storage was effectively inhibited by UF, which may be closely related to the higher protein solubility and lower oxidation degree of UF samples than AF and IF samples.

Benjakul et al. [46] suggested that MP gel strength was also affected by the stability of MP. Our previous results showed that fish MP treated with UF consistently had the highest thermal stability among all samples during frozen storage [10], which corresponded to the result of gel strength.

### 3.9. WHC of Fish Protein Gel

The WHC of the gel refers to the ability of gel network structure to hold water through protein–water interaction. Figure 6 shows that at the beginning of storage, the WHC of UF gel was obviously higher than that of IF and AF gel (*P* < 0.05). The WHC of UF and AF samples decreased significantly during the first 90 days of frozen storage and then decreased slowly. At 180 days of storage, the WHC of UF and AF gels decreased to 35.60% and 28.30%, respectively. The WHC of IF gels decreased rapidly in the first 60 days of frozen storage and slowed down at 60–90 days. The WHC of IF samples decreased to 31.19% at the end of storage. As shown in Figure 6, gel strength and gel WHC declined in a similar trend and the UF samples maintained the highest gel strength and gel WHC among all the samples. The change of WHC in the gel is closely related to the change of gel microstructure. The dense and uniform microstructure of the gel is conducive to the retention of water in the gel. Hu et al. [47] found that the unfolding of protein structure exposed hydrophobic groups, increased the degree of protein denaturation and decreased the WHC of the gel. Consistent with his findings, we also found that the unfolding of the protein structure of fish protein during storage increased hydrophobic groups, which reduced the WHC of the fish gel.

Wu, Hua, Lin and Xiao [48] thought that the WHC of the gel can be improved with increased protein solubility and decreased of particle size. The aggregation of proteins leads to a decrease in protein solubility, which prevents the proteins from fully unrolling during the heating phase at the beginning of gel formation, thus forming a weak gel and resulting in a decrease in gel WHC [49]. Of all the groups in this study, the UF samples always had the smallest particle size during storage; thus, the gel had the highest WHC.

### 3.10. Microstructure of Fish Protein Gel

The microstructure of gel is helpful to better understand the reason for the change of gel WHC and gel strength. Figure 7 shows that all gel samples showed different micro-morphologies after being frozen in different ways. The UF gels were characterized by small and uniform holes and the AF sample gel network had slightly larger holes with a loose distribution. With increasing storage time, the pores of the gel gradually increased and the shape became irregular. Some of the pores were connected to form larger pores. This could also explain the decrease in the gel WHC and gel strength. For samples at the same storage time, the gel pores of UF samples were always smaller than those of AF and IF samples. At the end of storage, the gel pores of AF samples were the largest and many small pores were connected to form large pores. The gel network structure was thus fractured. This may be because frozen storage led to protein denaturation and a decrease in protein solubility making the protein form a weak gel. In addition, the MP structure was disrupted during frozen storage and some active groups were exposed and oxidized resulting in the formation of intra- or intermolecular cross-links between proteins and the formation of large insoluble aggregates, which made proteins unable to unfold sufficiently during heating and thus they formed weaker gels [50]. Shenouda et al. [51] showed that the destruction of muscle proteins by ice crystals during frozen storage and the loss of moisture during thawing were possible reasons for the reduced gelation capacity.

## 4. Conclusions

The carbonyl and dityrosine content of UF group were lower than those of AF and IF groups, while the free amine content of UF was higher than that of AF and IF. This indicates that the UF inhibited the protein oxidation caused by frozen storage. In addition, the smaller particle size of UF samples further proves that the UF can inhibit the oxidative aggregation denaturing of proteins induced by frozen storage. Compared to the AF and IF protein gel, the UF protein gel had a higher *G*’, *G*”, gel WHC and gel strength indicating that the UF proteins can form better gels during frozen storage. The SEM image of the gel microstructure showed that UF protein gel was characterized by smaller and finer pores than other samples, which also proves that the UF protein formed a better gel. The water distribution result showed that the *T*_2b1_, *T*_2b2_, *T*_21_ and *T*_22_ relaxation time of UF gel were lower than those of IF and AF samples indicating that UF reduces the mobility of bound water, immobile water and free water of protein gel. This may be because UF increased the freezing speed and promote the formation of fine and uniform ice crystals. Frozen muscles were less damaged by ice crystals, cell contents (including endogenous proteases and water) leaked less and there was little change in the environment where proteins were located. Therefore, the oxidation of protein and the loss of protein gel properties during the frozen storage were inhibited by UF.

## Figures and Tables

**Figure 1 foods-10-00629-f001:**
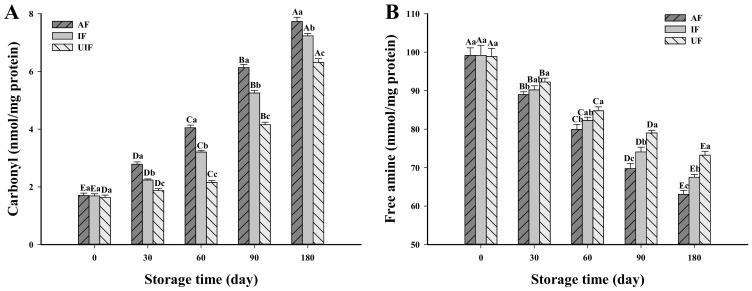
Change in carbonyl content (nmoL/mg protein) (**A**) and free amine content (mmoL/mg protein) (**B**) of fish myofibrillar protein frozen with different freezing methods (IF: immersion freezing; AF: air freezing; and UF: ultrasonic freezing) during storage. ^A–E^ indicate significant differences between different storage times in the same treatment group and ^a–c^ indicate significant differences between different frozen samples at the same storage time.

**Figure 2 foods-10-00629-f002:**
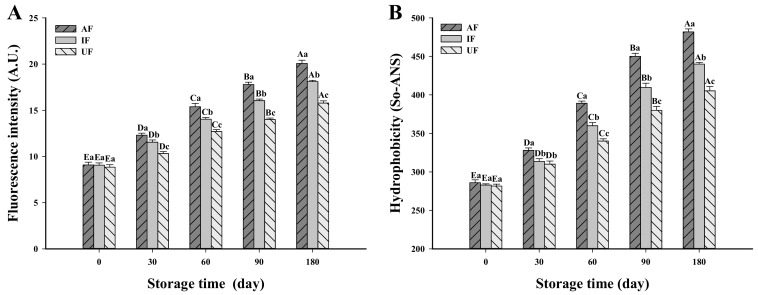
Change in dityrosine content (A.U.) (**A**) and hydrophobicity (So-ANS) (**B**) of fish myofibrillar protein frozen with different freezing methods (IF: immersion freezing; AF: air freezing; UF: ultrasonic freezing) during storage. ^A–E^ indicate significant differences between different storage times in the same treatment group and ^a–c^ indicate significant differences between different frozen samples at the same storage time.

**Figure 3 foods-10-00629-f003:**
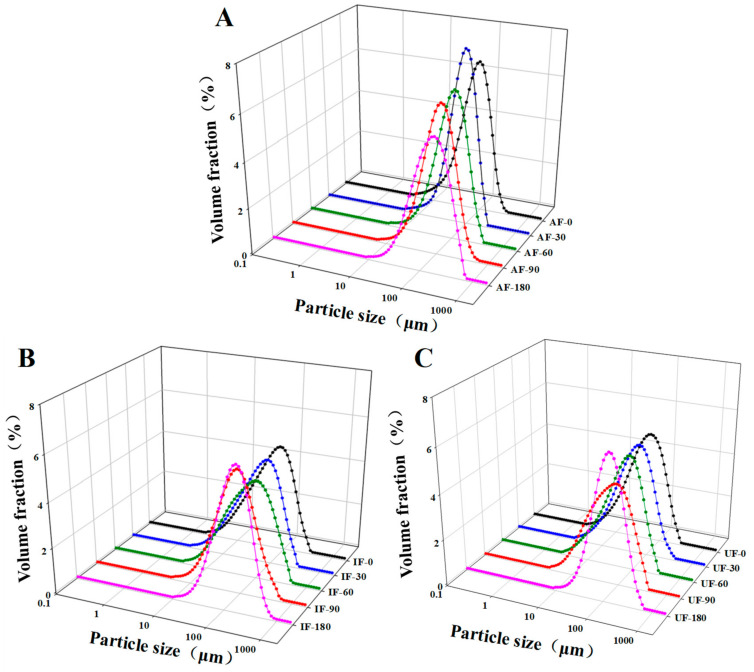
Change in particle size (μm) of fish protein frozen in different freezing methods (IF: immersion freezing (**B**); AF: air freezing (**A**); and UF: ultrasonic freezing (**C**) during storage.

**Figure 4 foods-10-00629-f004:**
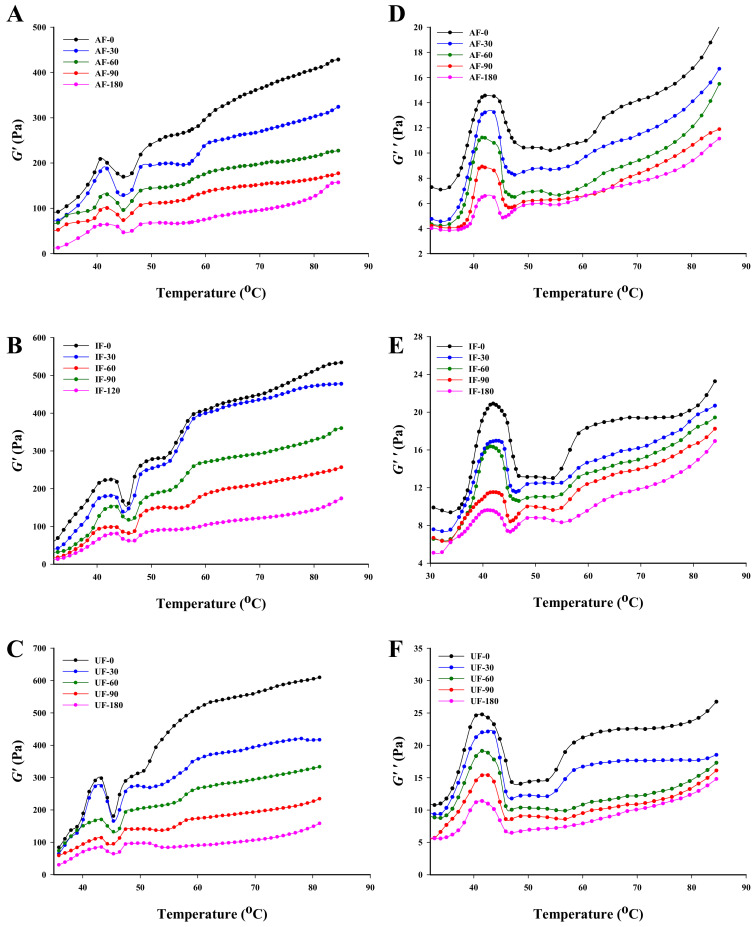
Change in *G*’ (**A**–**C**) and *G*” (**D**–**F**) of fish myofibrillar protein frozen in different freezing methods (IF: immersion freezing; AF: air freezing; and UF: ultrasonic freezing) during storage.

**Figure 5 foods-10-00629-f005:**
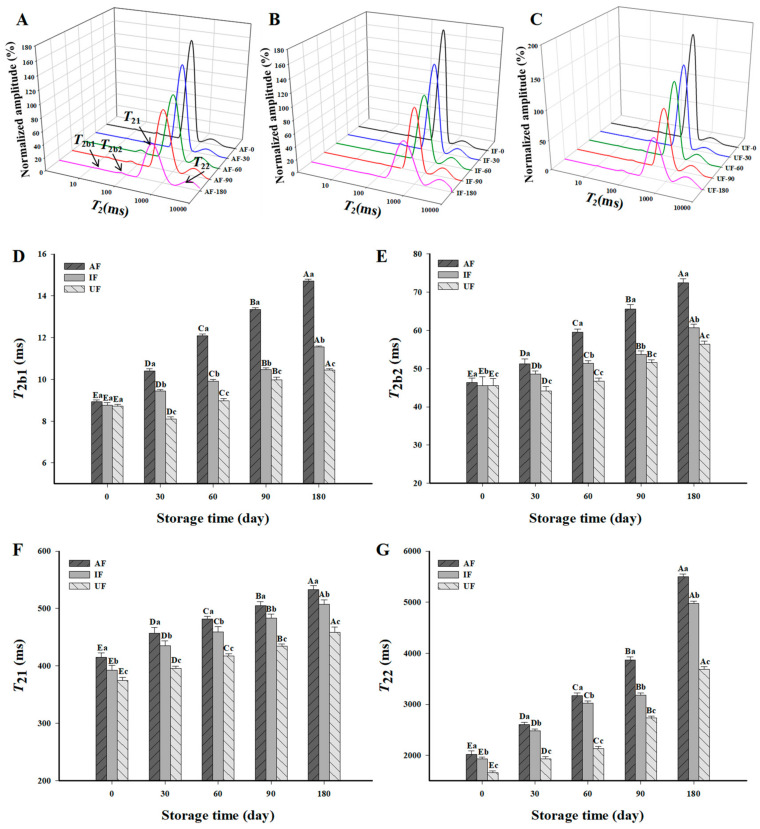
Change in low field nuclear magnetic resonance lines (**A**–**C**) and *T*_2_ relaxation times (ms) (**D**–**G**) of fish myofibrillar protein frozen in different freezing methods (IF: immersion freezing; AF: air freezing; UF: ultrasonic freezing) during storage. ^A–E^ indicate significant differences between different storage times in the same treatment group and ^a–c^ indicate significant differences between different frozen samples at the same storage time.

**Figure 6 foods-10-00629-f006:**
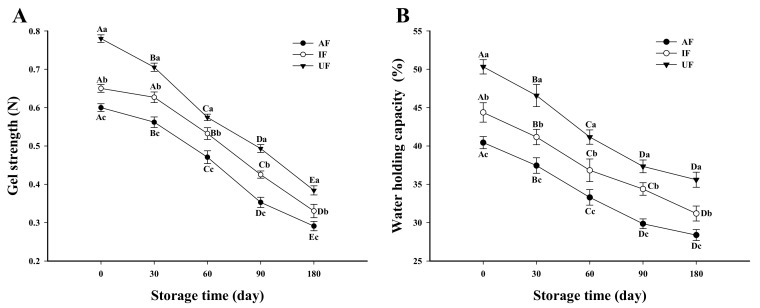
Change in protein gel strength (N) (**A**) and protein gel water holding capacity (%) (**B**) of fish myofibrillar protein frozen in different freezing methods (IF: immersion freezing; AF: air freezing; and UF: ultrasonic freezing) during storage. ^A–E^ indicate significant differences between different storage times in the same treatment group and ^a–c^ indicate significant differences between different frozen samples at the same storage time.

**Figure 7 foods-10-00629-f007:**
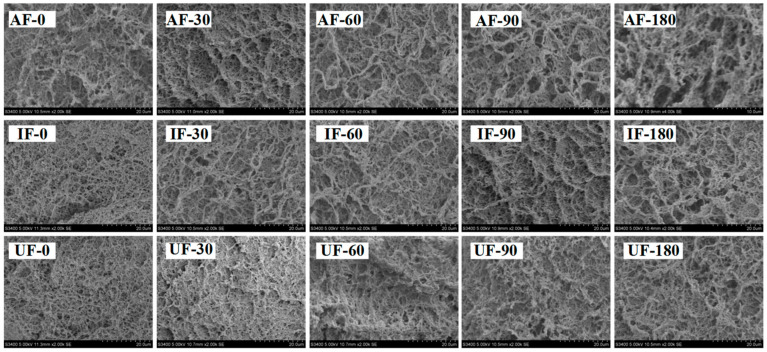
Scanning electron microscopy (SEM) images of myofibrillar protein gel of fish frozen with different freezing methods (IF: immersion freezing; AF: air freezing; and UF: ultrasonic freezing) during storage.

**Table 1 foods-10-00629-t001:** Changes in *D*_43_ (μm) of fish myofibrillar protein frozen with different freezing methods (IF: immersion freezing; AF: air freezing; and UF: ultrasonic freezing) during storage.

	Storage Time (Day)				
	0	30	60	90	180
AF	43.9 ± 0.7 ^Ea^	60.2 ± 0.7 ^Da^	67.7 ± 0.6 ^Ca^	81.6 ± 0.9 ^Ba^	113.4 ± 1.0 ^Aa^
IF	29.3 ± 0.9 ^Eb^	38.3 ± 0.9 ^Db^	42.5 ± 0.7 ^Cb^	54.5 ± 0.8 ^Bb^	109.5 ± 0.9 ^Ab^
UF	27.9 ± 0.6 ^Eb^	31.2 ± 0.4 ^Dc^	34.6 ± 0.5 ^Cc^	38.8 ± 0.7 ^Bc^	96.3 ± 0.8 ^Ac^

^A–E^ indicate significant differences between different storage times in the same treatment group and ^a–c^ indicate significant differences between different frozen samples at the same storage time.

**Table 2 foods-10-00629-t002:** Changes of relative area *A*_2_ (%) of different water amplitudes in protein gel with different freezing methods (IF: immersion freezing; AF: air freezing; and UF: ultrasonic freezing) during storage.

	Storage Time (Day)					
		0	30	60	90	180
*A* _2b1_	AF	0.32 ± 0.01 ^Ab^	0.32 ± 0.01 ^Aa^	0.28 ± 0.01 ^Ba^	0.25 ± 0.01 ^Ca^	0.24 ± 0.01 ^Ca^
	IF	0.34 ± 0.01 ^Ab^	0.33 ± 0.01 ^Aa^	0.29 ± 0.01 ^Ba^	0.25 ± 0.01 ^Ca^	0.24 ± 0.01 ^Ca^
	UF	0.38 ± 0.01 ^Aa^	0.34 ± 0.01 ^Ba^	0.29 ± 0.01 ^Ca^	0.26 ± 0.01 ^Da^	0.24 ± 0.01 ^Da^
*A* _2b1_	AF	0.33 ± 0.01 ^Ac^	0.31 ± 0.01 ^ABb^	0.29 ± 0.01 ^BCb^	0.26 ± 0.01 ^CDa^	0.25 ± 0.01 ^Da^
	IF	0.35 ± 0.01 ^Ab^	0.32 ± 0.01 ^Bb^	0.29 ± 0.01 ^Cb^	0.26 ± 0.01 ^Da^	0.26 ± 0.01 ^Da^
	UF	0.39 ± 0.01 ^Aa^	0.35 ± 0.01 ^Ba^	0.31 ± 0.01 ^Ca^	0.27 ± 0.01 ^Da^	0.26 ± 0.01 ^Ea^
*A* _21_	AF	92.62 ± 0.07 ^Ac^	90.87 ± 0.17 ^Bc^	87.13 ± 0.09 ^Cc^	85.83 ± 0.10 ^Dc^	81.4 ± 0.07 ^Ec^
	IF	93.00 ± 0.06 ^Ab^	92.25 ± 0.03 ^Bb^	88.09 ± 0.08 ^Cb^	86.44 ± 0.10 ^Db^	82.78 ± 0.09 ^Eb^
	UF	93.35 ± 0.04 ^Aa^	92.57 ± 0.09 ^Ba^	89.58 ± 0.07 ^Ca^	87.82 ± 0.13 ^Da^	84.75 ± 0.11 ^Ea^
*A* _22_	AF	6.73 ± 0.09 ^Ac^	8.41 ± 0.07 ^Bc^	12.33 ± 0.05 ^Cc^	13.69 ± 0.07 ^Dc^	18.11 ± 0.05 ^Ec^
	IF	6.31 ± 0.03 ^Ab^	7.11 ± 0.06 ^Bb^	11.32 ± 0.04 ^Cb^	13.03 ± 0.03 ^Db^	16.75 ± 0.08 ^Eb^
	UF	5.83 ± 0.04 ^Aa^	6.75 ± 0.07 ^Ba^	9.77 ± 0.09 ^Ca^	11.69 ± 0.03 ^Da^	14.72 ± 0.06 ^Ea^

^A–E^ indicate significant differences between different storage times in the same treatment group and ^a–c^ indicate significant differences between different frozen samples at the same storage time.

## Data Availability

No new data were created or analyzed in this study. Data sharing is not applicable to this article.

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
