# Peer review of "Ultrasonic Freezing Reduces Protein Oxidation and Myofibrillar Gel Quality Loss of Common Carp (Cyprinus carpio) during Long-Time Frozen Storage"

_foods, 2021, doi:10.3390/foods10030629_

Round 1

Reviewer 1 Report

In this paper the authors compare the impact of long-term frozen storage of common carp after ultrasonic freezing (UF), immersion freezing (IF) and air freezing (AF), on myofribrillar proteins (MP) oxidation and gel properties. Results show a clear positive impact of UF compared to IF and AF. This process would induce a faster freezing of samples associated to the formation of smaller ice crystals. According to the authors, the decrease of ice crystals would limit the breaking of fish tissue that would be at the origin of myofibrillar protein oxidation and denaturation.

Specific comments:

-The description of the surface hydrophobicity is incomplete, the treatment with ANS is not described and it is not mentioned that the signal measured is the fluorescence of the ANS-protein conjugate.

- The rheological properties were measured on MP solutions at 30 mg/ml along a temperature ramp from 30°C to 80°C at 1°C/min. The choice of this experimental protocol is not clear. Is it the thermal gelation of samples that is of interest for authors or the rheological properties of the gel samples? Why measurements are not directly performed on the MP gel or along the thermal history that is used to prepare gels? The MP gel investigated later (measuring water holding capacity, gel strength LF-NMR, SEM)  is prepared with a 40 mg/ml MP solution that is heated at 45°C for 15 min and at 80°C for 15 min.

Why a two-step heating process is chosen to prepare MP gels?

In addition, the authors should precise the frequency and the amplitude used to measure G’ and G”.

-l172: The sentence is not very clear. “….the samples were washed with 0.2M PBS three times and washed with ethanol (50%, 70%...100% respectively)”. Are samples washed with successive solvents composed with an increasing content of ethanol?

- l188: The value 9.25 nmol/mg does not correspond to the value presented in figure 1.

-The change of free amine content appears directly related to the carbonyl groups. Free amine and carbonyls contents results could be presented together in a same figure.

- l279: The relationship between the destruction of muscle tissues and unfolding degree of proteins is not clearly described. Is it related to the change of environment of proteins associated to the destruction of muscles?

-l301: The Mastersizer 2000 is not a dynamic light scattering apparatus; it is a granulometer that use laser diffraction principles.

-Table 1: There is not real interest to display both D43 and D32 values.

Please correct the written values with the correct number of digits in order to give one significant digit for error and display measured values with a precision with the same order of magnitude as the error : e.g.  43.87+/- 0.73 should be written 43.9+/-0.7.

-Figure 4: The peaks of G’ and G” around 42°C are not described and commented in the article. I suppose that it is related to the thermal denaturation of myofibrillar proteins, could you discuss it?

-I am not an expert in NMR, but I suppose that with the experimental results presented, the proportion of thightly bound water, immobilized water and free water could be deduced. Why isn’t it analyzed? It could be related to the gel microstructure and WHC. In addition, it seems that the distribution of T2 times display an enlargement of the T21 peak (at least) with ageing time since the maximum normalized intensities are decreasing with storage time. Is it correct? If yes, this kind of information could be also analyzed.

Minor comments:

-l184 “indexes” should be written “index”.

-Figure 1 B: The title of the Y-axis should be “Fluorescence intensity” instead of “Fluorescent intensity”.

-l271 The surface hydrophobicity value is written with too many digits compared to the error bars.

-l322 replace “more large” by “larger”

-l352 “causing by unfolding of protein structures” should be replaced by “causing unfolding of protein structures”

-l358 “the change of G” was similar to that of G”…” should be replaced by “the change of G” was similar to that of G’…”

-l374 “Fig. 3” should be replaced by “Fig.5”

-l472 the dot after “thus” should be removed.

Reviewer 2 Report

Article titled "Ultrasonic Freezing Reduces Protein Oxidation and Myofibrillar Gel Quality Loss of Common Carp (Cyprinus carpio) during Frozen Storage" could be of interest for Foods readers. It presents the effect of ultrasonic freezing (UF) on myofibrillar protein oxidation and gel properties
of common carp (Cyprinus carpio) during frozen storage with air freezing (AF) and immersion freezing (IF) as controls. In my opinion, all the chapters have been thoroughly described. In Introduction section all most important information has been carefully presented. The results have been shown clearly and appropriately graphically. The discussion is comprehensive and gives the possibility to compare the results obtained by Authors with other researches.

In my opinion abstract and conclusions need some corrections. The word "to have" has been used by Authors too often, what makes the text hard to be read -f.e. abstract - lines 16, 18, 20, 21; conclusions: lines 496, 500, 503, 504. In my opinion it should be replaced by other phrases, f.e. "was characterized by"etc.

Line 292 - The sentence"...the ultrasound used here was used..." is stylistically incorrect, please rearrange the sentence.

Author Response

Reviewer # 2

Article titled "Ultrasonic Freezing Reduces Protein Oxidation and Myofibrillar Gel Quality Loss of Common Carp (Cyprinus carpio) during Frozen Storage" could be of interest for Foods readers. It presents the effect of ultrasonic freezing (UF) on myofibrillar protein oxidation and gel properties of common carp (Cyprinus carpio) during frozen storage with air freezing (AF) and immersion freezing (IF) as controls. In my opinion, all the chapters have been thoroughly described. In Introduction section all most important information has been carefully presented. The results have been shown clearly and appropriately graphically. The discussion is comprehensive and gives the possibility to compare the results obtained by Authors with other researches.

Q1: In my opinion abstract and conclusions need some corrections. The word "to have" has been used by Authors too often, what makes the text hard to be read -f.e. abstract - lines 16, 18, 20, 21; conclusions: lines 496, 500, 503, 504. In my opinion it should be replaced by other phrases, f.e. "was characterized by"etc.

A1: Thank you for your advice. The abstract and the conclusions have been revised as follows:

Line 12-25: Abstract: The effect of ultrasonic freezing (UF) on myofibrillar protein oxidation and gel properties of common carp (Cyprinus carpio) during frozen storage were investigated with air freezing (AF) and immersion freezing (IF) as controls. The results showed that the carbonyl and dityrosine content of UF samples were lower, and the free amine content was higher than those of AF and IF samples during frozen storage indicating that UF inhibited protein oxidation caused by frozen storage. The particle size of UF sample was the smallest among all the groups indicating that UF inhibited the protein aggregation. The UF sample had higher G', G" value, gel strength, and gel water holding capacity than AF and IF groups showing that UF reduced the loss of protein gel properties. The gel microstructure showed that UF protein gel was characterized by smaller and finer pores than other samples, which further proves that UF inhibited loss of gel properties during frozen storage. The UF sample had shorter T2 transition time than other samples demonstrating that UF decreased the mobility of water. In general, UF is an effective method to reduce protein oxidation and gel properties loss caused by frozen storage.

Line 560-576: 4. Conclusions

The carbonyl and dityrosine content of UF group were lower than those of AF and IF groups, while the free amine content of UF was higher than that of AF and IF. This indicates that the UF inhibited the protein oxidation caused by frozen storage. In addition, the smaller particle size of UF samples further proves that the UF can inhibit the oxidative aggregation denaturing of proteins induced by frozen storage. Compared to the AF and IF protein gel, the UF protein gel had a higher G', G", gel WHC, and gel strength indicating that the UF proteins can form better gels during frozen storage. The SEM image of the gel microstructure showed that UF protein gel was characterized by smaller and finer pores than other samples, which also proves that the UF protein formed a better gel. The water distribution result showed that the T2b1, T2b2, T21, and T22 relaxation time of UF gel were lower than those of IF and AF samples indicating that UF reduces the mobility of bound water, immobile water, and free water of protein gel. Overall, the UF inhibits protein oxidation and deterioration of gel properties due to frozen storage.

Q2: Line 292 - The sentence"...the ultrasound used here was used..." is stylistically incorrect, please rearrange the sentence.

A2: Thank you. The sentence has been revised as “In addition, in the present study, the ultrasound was used during the freezing process ...” (Line 316)

Reviewer 3 Report

  This paper has been very well-organized and English is very clear.  The results are giving an interesting and valuable information.  However, there are some problems and flaws in presentation and discussion.  I hope that my comments are very useful for the improvement of this research.

Comments

  • Title: I think the title " Ultrasonic Freezing Reduces Protein Oxidation and Myofibrillar Gel Quality Loss of Common Carp (Cyprinus carpio) during Long-time Frozen Storage " will have more impact.
  • Abstract: Since ultrasonic freezing (UF) is not a common method, it is helpful to briefly show this principle and feature in the abstract.
  • Abstract: Does "smallest particle size" mean "smallest particle size of MP"? Please correct it.
  • I need to explain what G’ and G’’ are.
  • Introduction: Ultrasonic freezing (UF) is not a common method, so please explain this principle and characteristics in detail.
  • Introduction: Please show the reason why authors chose carp as the subject of this experiment.
  • Introduction: Please explain why authors chose this storage period (maximum 180 days).
  • Materials and Methods: There is no description of the thawing method for frozen samples, so please describe it. This is a very important point.
  • Result and Discussion: In Table 1, please consider a significant number, and then modify values.
  • Result and Discussion: In Figure 6, gel strength and WHC decreased almost linearly with any freezing method. The difference between day 0 of each group was maintained. Does the parameters have nothing to do with the size of ice crystals depending on the freezing method? Please add your thoughts.
  • Result and Discussion: How does UF reduce the size of ice crystals when freezing? Please describe if you have any information.
  • Conclusion: The advantages of UF are shown in the measuring parameters of this experiment. Why did UF advantage from these parameters over other freezing methods (AF and IF)? Please discuss this point.

Author Response

Reviewer # 3

This paper has been very well-organized and English is very clear. The results are giving an interesting and avaluable information. However, there are some problems and flaws in presentation and discussion. I hope that my comments are very useful for the improvement of this research.

Comments

Q1: Title: I think the title " Ultrasonic Freezing Reduces Protein Oxidation and Myofibrillar Gel Quality Loss of Common Carp (Cyprinus carpio) during Long-time Frozen Storage " will have more impact.

A1: Thank you for your suggestion. The title has been revised as “Ultrasonic Freezing Reduces Protein Oxidation and Myofibrillar Gel Quality Loss of Common Carp (Cyprinus carpio) during Long-time Frozen Storage”. (Line 4)

Q2: Abstract: Since ultrasonic freezing (UF) is not a common method, it is helpful to briefly show this principle and feature in the abstract.

A2: Thank you for your comments. The following sentences have been added to the abstract:

Ultrasonic freezing (UF) is an effective method to increase the freezing speed and improve the quality of frozen food. (Line 12-13)

Q3: Abstract: Does "smallest particle size" mean "smallest particle size of MP"? Please correct it.

A3: Thank you. The "smallest particle size" has been changed to "The particle size of UF MP was the smallest"? (Line 18)

Q4: I need to explain what G’ and G’’ are.

A4: Thank you for your advice. The following sentence has been added to the manuscript:

The G' reflects the elastic solid-state behaviour of the sample, and the G'' measures the viscous response of the sample. (Line 364-365)

Q5: Introduction: Ultrasonic freezing (UF) is not a common method, so please explain this principle and characteristics in detail.

A5: Thank you for your suggestion. The following sentences have been added to the manuscript:

The mechanisms may be as follows. On the one hand, ultrasonic waves can produce cavitation bubbles, which can be used as crystal nuclei to promote the formation of ice crystals. On the other hand, micro-jet generated by ultrasonic wave can break large ice crystals into small ice crystals, which can be used as crystal nuclei to promote recrystallization of ice crystals [5-6]. (Line 42-47)

Cited references:

[5] Tian, Y.; Chen, Z.; Zhu, Z.; Sun, D. W. Effects of tissue pre-degassing followed by ultrasound-assisted freezing on freezing efficiency and quality attributes of radishes. Ultrason. Sonochem. 2020, 67, 105162. https://doi.org/10.1016/j.ultsonch.2020.105162

[6] Li, Y.; Zhang, Y.; Liu, X.; Wang, H.; Zhang, H. Effect of ultrasound-assisted freezing on the textural characteristics of dough and the structural characterization of wheat gluten. J. Food Sci. Tech. Mys. 2019, 56(7), 3380-3390. https://doi.org/10.1007/s13197-019-03822-6

Q6: Introduction: Please show the reason why authors chose carp as the subject of this experiment.

A6: Thank you for your advice. The following sentences have been added to the manuscript:

  1. Introduction

Common carp (Cyprinus carpio) is the most important freshwater fish in the world, especially in China. There were about 3.01 million tons of carp were farmed in 2019, accounting for about 70% of total freshwater fish production in China. It is rich in protein, unsaturated fatty acids, and trace elements. They are a common food that easily spoilage and deteriorates; therefore, measures should be taken to better preserve fish [1]. (Line 29-31)

Q7: Introduction: Please explain why authors chose this storage period (maximum 180 days).

A7: Thank you for your advice. The following sentence has been addded to the manuscript:

The storage period of frozen fish was about 90-180 d, the protein and quality of fish changed significantly after long term frozen storage (180 d). (Line 33-35)

Q8: Materials and Methods: There is no description of the thawing method for frozen samples, so please describe it. This is a very important point.

A8: The frozen samples in the present study were thawed in a refrigerator (4 °C). The thawing method has been added to the manuscript as follow:

The frozen samples were thawed in a refrigerator (4 °C) to a central temperature of 4 °C. Then the fish meat was removed from the fish piece and minced. (Line 93-94)

Q9: Result and Discussion: In Table 1, please consider a significant number, and then modify values.

A9: Thank you for your advice. And the Table 1 has been revised as follow:

Table 1. Changes in D43 (μm) of myofibrillar protein of fish myofibrillar protein frozen with different freezing methods (IF: immersion freezing; AF: air freezing; and UF: ultrasound-assisted immersion freezing) during storage.

Storage time (d)

0

30

60

90

180

AF

43.9±0.7Ea

60.2±0.7Da

67.7±0.6Ca

81.6±0.9Ba

113.4±1.0Aa

IF

29.3±0.9Eb

38.3±0.9Db

42.5±0.7Cb

54.5±0.8Bb

109.5±0.9Ab

UF

27.9±0.6Eb

31.2±0.4Dc

34.6±0.5Cc

38.8±0.7Bc

96.3±0.8Ac

A-E indicate significant differences between different storage times in the same treatment group, and a-c indicate significant differences between different frozen samples at the same storage time. (Line 357-360)

Q10: Result and Discussion: In Figure 6, gel strength and WHC decreased almost linearly with any freezing method. The difference between day 0 of each group was maintained. Does the parameters have nothing to do with the size of ice crystals depending on the freezing method? Please add your thoughts.

A10: The degree of decline in gel strength and gel water holding capacity among different freezing groups was indeed almost consistent. We include the following discussion in the manuscript:

As shown in Fig. 6, gel strength and gel water holding capacity declined in a similar trend, and the UF samples maintained the highest gel strength and gel water holding capacity among all the samples. (Line 521-523)

Q11: Result and Discussion: How does UF reduce the size of ice crystals when freezing? Please describe if you have any information.

A11: Thank you for your suggestion. The following sentences have been added to the manuscript:

The cavitation bubbles generated by ultrasound can act as crystal nuclei to promote the formation of ice crystals. In addition, the micro-jet produced by ultrasound can break the large ice crystals into small ice crystals, which can be used as crystal nuclei and promote the recrystallization of ice crystals [9-10]. Therefore, UF can promote the generation of fine and uniform ice crystals, and inhibit the protein denaturation produced by freezing. (Line 317-321)

Q12: Conclusion: The advantages of UF are shown in the measuring parameters of this experiment. Why did UF advantage from these parameters over other freezing methods (AF and IF)? Please discuss this point.

A12: Thank you for your advice. The following sentences have been added to the manuscript:

This may be because UF increased the freezing rate and promote the formation of fine and uniform ice crystals. Frozen muscles were less damaged by ice crystals, cell contents (including endogenous proteases and water) leaked less, and there was little change in the environment where proteins were located. Therefore, the oxidation of protein and the loss of protein gel properties during the frozen storage were inhibited by UF. (Line 571-576)

Round 2

Reviewer 1 Report

Reviewers comments were well taken into account and the manuscript is very  improved. As a consequence, I accept the manuscript in the present form.